# Effects of Different Sources of Iron on Growth Performance, Immunity, and Intestinal Barrier Functions in Weaned Pigs



**Limei Sun** [1,2], **Bing Yu** [1,2], **Yuheng Luo** [1,2], **Ping Zheng** [1,2], **Zhiqing Huang** [1,2], **Jie Yu** [1,2], **Xiangbing Mao** [1,2], **Hui Yan** [1,2], **Shaohui Wang** [3] **and Jun He** [1,2,*]

1    Animal Nutrition Institute, Key Laboratory for Animal Disease-Resistance Nutrition of Sichuan Province, Sichuan Agricultural University, Chengdu 611130, China
2    Key Laboratory of Animal Disease-Resistant Nutrition, Chengdu 611130, China
3    Jiangsu Shuxing Biotechnology Co., Ltd., Chengdu 610207, China
*    Correspondence: hejun8067@163.com; Tel./Fax: +86-28-86293067

**Abstract:** To explore the effect of different sources of iron on growth performance and intestinal health, 24 weaned pigs were randomly allocated to three groups and fed with a basal diet (BD) or BD containing 100 mg/kg ferrous sulfate (FS) or ferrous glycine (FG). The trial lasted for 21 d, and blood and tissue were collected for analysis. Results showed that FG significantly decreased the feed-to-gain ratio and increased the iron content in the liver and tibia ($p < 0.05$). Both FS and FG elevated bladder and fecal iron content and significantly elevated the contents of red blood cells, hemoglobin (HGB), and ferritin in the blood ($p < 0.05$). FG supplementation increased the serum concentrations of immunoglobulin (Ig) G and IgM, but decreased the concentrations of D-lactate and endotoxin ($p < 0.05$). Interestingly, FG significantly increased the villus height and the ratio of villus height to crypt depth (V/C) in the duodenum and ileum ($p < 0.05$). FG supplementation also increased the abundance of tight-junction protein ZO-1 but significantly decreased the rate of apoptosis in the jejunum ($p < 0.05$). Moreover, the activities of jejunal sucrase, maltase, and catalase (CAT) in the FG group were higher than that in other groups ($p < 0.05$). Importantly, FG not only elevated the expression levels of cationic amino acid transporter-1 (CAT1) in the duodenal and jejunum but also elevated the expression levels of glucose transporter-2 (GLUT2) and sodium/glucose co-transporter 1 (SGLT1) in the ileum ($p < 0.05$). These results indicated that appropriate iron supplementation is beneficial to piglet intestinal health by enhancing immunity and improving antioxidant capacity, and FG may serve as an efficient substitute for conventionally used iron sources.

**Keywords:** iron; nutrition; organic iron; immunity; intestinal health; weaned pigs

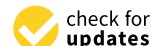



## 1. Introduction

Iron is one of the most important trace elements for animals, as it has been implicated in numerous vital biologic processes, such as the oxygen transport, DNA biosynthesis, and energy metabolism [1,2]. Moreover, the development of the immune system for mammals (including pigs) also depends on the availability of iron [3,4]. In pig production, iron deficiency is a common nutritional disease for neonatal pigs, as their iron storage in the body exhausts quickly and the breast milk usually cannot provide sufficient iron for the pigs [5]. Previous studies indicated that iron is a cofactor of peroxidases such as catalase and myeloperoxidases, which protect cells against the toxic effects of hydrogen peroxide [6,7]. Iron deficiency impairs the antioxidant system, which leads to oxidative stress and subsequently stimulates apoptosis of the intestinal epithelial cells in weaning pigs [6,8,9]. Moreover, previous research has shown that excess reactive oxygen species may increase the mucosal permeability, which makes the intestine more susceptible to toxin infection [10]. Importantly, iron deficiency usually leads to malnutrition, and severe iron deficiency even causes iron deficiency anemia or related dysfunction [11,12]. Dietary iron

supplementation has been looked at as an efficient avenue to prevent iron deficiency in animals [13]. However, the bioavailability of different forms of iron may vary widely [13,14]. For instance, inorganic iron, such as ferrous sulfate and ferrous phosphate, has long been used as a favorite iron source for pig production. However, the inorganic irons have poor biological availability and may react with other nutrients in the gut. In recent years, organic iron supplements, such as ferric fumarate and ferric citrate, have attracted considerable research interest because of their elevated absorption rates [15]. For instance, piglets fed with ferric citrate have improved Fe digestibility and oxidative status compared with pigs fed with inorganic iron [16]. Earlier studies in pigs also demonstrated that ferric fumarate was more safe and efficacious than inorganic iron [17].

Ferrous glycine (FG) is a novel organic iron source that is a chelate of amino acid and iron. As compared to other iron source, FG has many advantages such as few irritants to the gastrointestinal tract and improved bioavailability [18–20]. Previous studies indicated that dietary FG supplementation had a significant positive effect on the hematological and biochemical parameters of broilers' blood [21]. In addition, studies involving pigs indicated that glycine-chelated iron was well absorbed and utilized [22]. In contrast, a number of studies showed a positive effect of FG in animal production. However, the mechanism behind the FG-regulated biological events still remains unclear. Moreover, the effect of FG may vary depending on animal species, sex, dose, and the physiological stage of animals.

The aim of this study was to explore the effect of different iron sources (FG vs. FS) on growth performance, immunity, intestinal morphology, and health in weaned piglets. This study will assist in the rational selection of dietary iron sources for animals.

## 2. Materials and Methods

### 2.1. Animal Trial

All the procedures used in the animal experiment were approved by the Institutional Animal Care and Use Committee of Sichuan Agricultural University. Twenty-four weaned boars (Large White × Landrace × Duroc) with an average initial body weight of $9.76 \pm 0.13$ kg were randomly assigned to one of the following dietary treatments: BD group (basal diet with no extra iron addition), FS group (basal diet with iron compensation by $FeSO_4.7H_2O$, $FeSO_4.7H_2O$ was bought from China National Pharmaceutical Group Corporation, purity 99%), and FG group (basal diet with iron compensation by Fe-Gly, Fe-Gly was kindly provided by Jiangsu Shuxing Biotechnology Co., Ltd. Fe $\geq$ 17.0%; chelate ratio $\geq$90%). The basal diet is an iron-deficient diet (25.8 mg/kg), which was formulated to meet the nutrient requirements (excluding the iron) recommended by the National Research Council 2012 [23]. To reduce the basal iron content in the diet, feedstuffs with high iron content (e.g., soybean meal) were not used in the diet (Table S1). The final iron content in the FS and FG groups was 100 mg/kg (element). Each treatment consisted of 8 pigs, and pigs were allocated to each treatment from the same littermates. Each pig was individually housed in a metabolic cage (0.7 m × 1.5 m) and were allowed access to food and water ad libitum with room temperature maintained at 25–28 °C and relatively controlled humidity (55–65%). The trial lasted for 21 d after 7 d adaptive period to the treatment diets. On day 22, after 12 h fasting, the BW of each pig was measured. The feed intake of each pig was measured daily, and the feed-to-gain ratio (G:F) of each pig was calculated according to the average daily gain (ADG) and average daily feed intake (ADFI).

### 2.2. Sample Collection and Treatment

Fresh fecal samples were collected during d 18–21 of the trial. Immediately after defecation, fresh feces of each pen were collected into their own sealed plastic bags, and then 10 mL of 10% $H_2SO_4$ solution was evenly added to 100 g of feces to fix fecal nitrogen. On d 22, the bloods were obtained by jugular vein puncture, and serum samples were obtained after centrifuged at $3500\times g$ at 4 °C for 15 min and then frozen at $-20$ °C for analysis. The whole blood samples were obtained by EDTA anticoagulant and sent to the Ya'an People's Hospital for hematological analysis. Then, pigs were slaughtered, and the

tissue (liver, gallbladder, and kidney) and tibia samples were immediately collected and stored at −80 °C. Separate parts of the duodenum, jejunum, and ileum (approximately 4 cm of each tissue) were immediately and slowly flushed with cold phosphate buffered before being fixed in 4% paraformaldehyde solution for morphological analyses and immunofluorescence. Moreover, the mucosa samples were scraped with a scalpel blade from duodenum, jejunum, and ileum segments and preserved at −80 °C until analysis.

### 2.3. Iron Content in Diet, Fecal, and Tissue Samples

The iron content in the diet, fecal samples, tibia, and tissues (liver, gallbladder, and kidney) was determined using a method described by Shelton and Southern [24]. Samples were dried at 100 °C for 24 h and ashed for 10 h (diet, liver, gallbladder, kidney, and feces) or 36 h (tibia) at 550 °C. The ashed samples were dissolved in a nitric acid-perchloric acid mixture (1:1) and diluted with deionized water for analysis of iron [25]. Contents of Fe were measured with flame atomic absorption spectrophotometry (AA-6300, Shimadzu Corp., Tokyo, Japan).

### 2.4. Hematological Parameters Analysis

The whole blood samples were sent to the Ya'an People's Hospital for hematological measurements, including the total red blood cell number (RBC), hemoglobin (HGB), hematocrit (HCT), mean corpuscular volume (MCV), and total platelet number (PLT), which were determined by the BC-5000VET automatic vet hematology analyzer for animals (Mindray, Guangdong, China). The enzyme-linked immunosorbent assay (ELISA) kits purchased from Jiangsu Enzyme-linked Biotechnology Co., Ltd. (Jiangsu, China) were used for the detection of serum ferritin (MM77617O1) and transferrin (MM32623O1). All procedures referred to the instructions of the kits.

### 2.5. Serum Parameter Analysis

The ELISA kits purchased from Jiangsu Enzyme-linked Biotechnology Co., Ltd. (Jiangsu, China) were used to detect the insulin-like growth factor-1 (Kit MM-0389O1), immunoglobulin (Ig) G (Porcine IgG ELISA Kit MM-0403O1), immunoglobulin (Ig) M (MM-0402O1), immunoglobulin (Ig) A (MM-0905O1), D-lactate (MM33732O1), endotoxin (MM-36368O1), and diamine oxidase (MM-0438O1). All procedures referred to the instructions of the kits.

The content of blood urea nitrogen (BUN) and D-xylose was respectively measured using the blood urea nitrogen assay kit (C013-2-1) and D-xylose assay kit (A035-1-1) purchased from Nanjing Jiancheng Biotechnology Co., Ltd. (Nanjing, China). All procedures referred to the instructions of the kits, respectively.

### 2.6. The Intestinal Antioxidant Capacity Analysis

The commercial kits purchased from Nanjing Jiancheng Biotechnology Co., Ltd. (Nanjing, China) were used for the detection of catalase (The CAT Assay kit A007-1-1), malondialdehyde (The MDA Assay kit A003-1-2), glitathione peroxidase (The GSH-PX Assay kit A005-1-2), total superoxide dismutase (The T-SOD assay kit A001-1-2), and antioxidant capacity (The T-AOC assay kit A015-1-2) in the small intestinal mucosa (duodenum, jejunum, and ileum). All procedures referred to the instructions of the kits, respectively.

### 2.7. The Intestinal Morphology Measurement

The fixed duodenum, jejunum, and ileum samples were removed from the 4% paraformaldehyde solution for dressing. The samples were dehydrated (alcohol), transparent (xylene), embedded (paraffin), and sectioned, then stained with hematoxylin and eosin (H&E) and sealed with neutral gum for histomorphological examination. An Eclipse CI-L photo microscope (Nikon, Japan) was used to select the target area of tissues for 40× imaging. During imaging, tissues filled the whole field of vision as far as possible to ensure the consistent background light of each photo. After the imaging was completed, image-Pro

Plus 6.0 analysis software was used to measure the height of 5 intact villi in each section with mm as the standard unit. Five crypt depths were calculated, and the average value was calculated.

### 2.8. The Immunofluorescence Measurement

We determined the distribution of ZO-1 protein in jejunal tissues by immunofluorescence. Firstly, the samples fixed with 4% paraformaldehyde were flushed in PBS before incubated with 1 mol/L ethylene diamine tetraacetic acid (EDTA, pH 9.0, Gooddbio Technology Co., Ltd., Wuhan, China) for antigen retrieval. Secondly, the 3% bovine serum albumin was used to block the tissue sections before incubating it with rabbit anti-ZO-1 polyclonal antibody all night at 4 °C. The next steps refer to our previous research [26].

### 2.9. The Flow Cytometry Assays

A section of jejunal tissues was determined for apoptosis by flow cytometry. Firstly, the jejunum tissue was rinsed with ice phosphate buffer, then applied to the ice pack. Secondly, we hung off the mucosal cells with a slide, cut the tissue block with scissors, added the right amount of RPMI 1640 medium (Hyclone, Logan, UT, America), and then placed it in a new tube before mixing the cells. The next steps are consistent with our previous research [26]. Finally, samples were detected by CytoFLEX flow cytometry (Backman, Brea, CA, America) with CytExpert software (Backman, Brea, CA, America).

### 2.10. Digestive Enzyme Activity

The frozen duodenal, jejunal, and ileal mucosa were homogenized in chilled saline at a ratio of 1:9 ($w/v$) for 15 min. The homogenate was centrifuged at $3500 \times g$ for 10 min at 4 °C, and the supernatant was used to determine the enzyme activities, including the lactase, sucrase, and maltase. The three enzyme activities were respectively determined using the sucrase assay kit (A082-2-1), lactase assay kit (A082-1-1), and maltase assay kit (A082-3-1) purchased from Nanjing Jiancheng Biotechnology Co., Ltd. (Nanjing, China). Enzyme activity was defined as hydrolysis of 1 mol of the substrate per mg of protein tissue per minute under the condition of 37 °C, pH = 6.0.

### 2.11. RNA Isolation, Reverse Transcription, and Real-Time Quantitative PCR

The frozen jejunal and ileal mucosal samples (approximately 0.1 g) were homogenized in 1 mL Trizol reagent (TaKaRa, Dalian, China), and total RNA was extracted according to the manufacturer's instructions. Then, reverse transcription was performed using the PrimeScript RT kit and gDNA Eraser (TaKaRa, Dalian, China) to obtain all samples according to the manufacturer's instructions. Quantitative real-time PCR was performed on a Q5 RealTime PCR detection system to analyze the expression levels of divalent metal transporter-1 (DMT1), peptide transporter-1 (PePT1), cationic amino acid transporter-1 (CAT1), zinc transporter-1 (ZnT1), Na+-dependent glucose transporter-1 (SGLT1), and glucose transporter-2 (GLUT2), which are presented in Table S2.

β-actin was selected as the reference gene for transcription. Primers for specific genes (see Table S2 for primer sequences) were synthesized commercially by Shenggong Bioengineering (Shanghai, China). The specific experimental approach was adopted from our previous report [27].

### 2.12. Statistical Analysis

The data were analyzed using a one-way analysis of variance (ANOVA) of SPSS 24.0 (SPSS, Inc., Chicago, IL, USA). Differences among the treatments were estimated using the Student–Newman–Keuls test multiple comparisons, and values were expressed as means with their standard errors (means ± SEM). The differences were considered significant at $p$-values < 0.05.

### 3. Results

*3.1. Effect of Different Iron Sources on Growth Performances in Weaned Pigs*

No difference was observed in the initial body (IBW) and average daily gain (ADG) among the three groups (Table 1). However, dietary FG supplementation decreased the feed-to-gain ratio ($p < 0.05$). As compared to the BD group, both the FS and FG supplementation increased the iron content in the gall bladder and fecal samples ($p < 0.05$). Dietary FG supplementation also elevated the iron content in the liver and tibia when compared with the BD group ($p < 0.05$).

**Table 1.** Effect of different iron sources on growth performances and iron deposition in weaned pigs.

| ITEM | Treatments | | | SEM | *p*-Value |
|---|---|---|---|---|---|
| | BD | FS | FG | | |
| IBW (kg) | 9.75 | 9.77 | 9.76 | 0.13 | 1.00 |
| ADG (g/d) | 227.56 | 248.23 | 236.83 | 8.62 | 0.62 |
| ADFI (g/d) | 475.83 [a] | 480.57 [a] | 406.60 [b] | 65.81 | 0.07 |
| FBW (kg) | 14.53 | 14.73 | 14.23 | 0.24 | 0.33 |
| F:G | 2.12 [a] | 2.01 [a] | 1.73 [b] | 0.06 | 0.02 |
| Liver (mg/kg) | 27.67 [b] | 45.18 [ab] | 53.75 [a] | 4.45 | 0.03 |
| Gall bladder (mg/kg) | 6.52 [b] | 10.79 [a] | 10.93 [a] | 0.68 | 0.01 |
| Kidney (mg/kg) | 16.83 | 19.40 | 17.24 | 0.95 | 0.51 |
| Tibia (mg/kg) | 180.00 [b] | 207.14 [ab] | 241.80 [a] | 10.16 | 0.04 |
| Fecal (mg/kg) | 669.14 [b] | 1454.71 [a] | 1347.57 [a] | 82.34 | 0.01 |

IBW, initial body weight; FBW, final body weight; ADG, average daily gain; ADFI, average daily feed intake; F:G, feed-to-gain ratio. Within a row, mean values of different letter superscripts were significantly different ($p < 0.05$). Mean and total SEM are listed in separate columns (n = 8). BD, basal diet with no iron supplementation; FS, basal diet with $FeSO_4$ (containing 100 mg Fe/kg) supplementation; FG, basal diet with Fe-Gly (containing 100 mg Fe/kg) supplementation.

*3.2. Effect of Different Iron Sources on Hematological Parameters in Weaned Pigs*

As compared to the BD group, FG and FS supplementation increased the count of RBC, HGB, and HCT ($p < 0.05$). Moreover, the serum concentration of ferritin increased in weaned pigs by FG and FS addition ($p < 0.05$). There were no differences observed in other parameters between the FG and FS groups (Table 2).

**Table 2.** Effect of different iron sources on hematological parameters in weaned pigs.

| ITEM | Treatments | | | SEM | *p*-Value |
|---|---|---|---|---|---|
| | BD | FS | FG | | |
| RBC ($10^{12}$/L) | 6.33 [b] | 7.07 [a] | 6.86 [a] | 0.10 | 0.00 |
| HGB (g/L) | 100.33 [b] | 110.38 [a] | 109.83 [a] | 1.85 | 0.04 |
| HCT (%) | 31.52 [b] | 35.56 [a] | 35.05 [a] | 0.66 | 0.02 |
| MCV (fL) | 49.87 | 50.29 | 51.25 | 0.83 | 0.82 |
| PLT ($10^9$/L) | 492.67 [b] | 662.50 [a] | 523.17 [ab] | 33.76 | 0.07 |
| Ferritin (ng/mL) | 134.40 [b] | 168.39 [a] | 167.52 [a] | 6.27 | 0.03 |
| Transferrin (μg/mL) | 16.83 | 16.03 | 16.25 | 0.28 | 0.65 |

RBC, red blood cell; HGB, hemoglobin; HCT, hematocrit; MCV, mean corpuscular volume; PLT, total platelet number. Within a row, mean values of different letter superscripts were significantly different ($p < 0.05$). Mean and total SEM are listed in separate columns (n = 8). BD, basal diet with no iron supplementation; FS, basal diet with $FeSO_4$ (containing 100 mg Fe/kg) supplementation; FG, basal diet with Fe-Gly (containing 100 mg Fe/kg) supplementation.

*3.3. Effect of Different Iron Sources on Serum Parameter in Weaned Pigs*

As compared to the FS and BD groups, FG supplementation increased the serum concentrations of IgG and IgM (Table 3, $p < 0.05$). Moreover, the serum concentrations of D-lactate and endotoxin were reduced by FG addition ($p < 0.05$).

**Table 3.** Effect of different iron sources on serum parameters in weaned pigs.

| ITEM | Treatments | | | SEM | *p*-Value |
|---|---|---|---|---|---|
| | BD | FS | FG | | |
| IGF-1 (μg/L) | 34.93 | 36.34 | 37.02 | 0.57 | 0.32 |
| IgA (μg/mL) | 22.37 | 21.32 | 31.23 | 2.84 | 0.29 |
| IgG (μg/mL) | 53.10 [b] | 68.21 [ab] | 82.50 [a] | 4.41 | 0.02 |
| IgM (μg/mL) | 20.33 [b] | 22.80 [b] | 30.33 [a] | 1.52 | 0.01 |
| DAO (pg/mL) | 203.66 | 214.93 | 188.87 | 11.12 | 0.65 |
| D-lactate (μg/L) | 499.44 [a] | 483.83 [a] | 383.31 [b] | 17.12 | 0.01 |
| ET (pg/mL) | 374.04 [a] | 334.99 [b] | 242.62 [c] | 13.15 | 0.00 |
| BUN (mmol/L) | 5.88 | 5.82 | 5.37 | 0.20 | 0.55 |
| D-xylose (mmol/L) | 1.20 | 1.28 | 1.32 | 0.06 | 0.68 |

IGF-1, insulin-like growth factor 1; IgA, immunoglobulins A; IgG, immunoglobulins G; IgM, immunoglobulins M; DAO, diamine oxidase; BUN, blood urea nitrogen; ET, endotoxin. Within a row, mean values of different letter superscripts were significantly different ($p < 0.05$). Mean and total SEM are listed in separate columns (n = 8). BD, basal diet with no iron supplementation; FS, basal diet with FeSO$_4$ (containing 100 mg Fe/kg) supplementation; FG, basal diet with Fe-Gly (containing 100 mg Fe/kg) supplementation.

*3.4. Effect of Different Iron Sources on Intestinal Morphology and Mucosal Enzyme Activity in Weaned Pigs*

The intestinal morphology is shown in Figure 1. As compared to the BD group, the integration of intestinal morphology in the FG group was higher than that in the BD group. In addition, the morphology of jejunal vills in the FG group was more complete than that in the FS group. Both FG and FS supplementation significantly increased the villus height and the ratio of V/C in the duodenum (Table 4). Dietary FG supplementation also elevated the villus height and the ratio of V/C in the ileum ($p < 0.05$). Both the FG and FS elevated the sucrase activity in the jejunal mucosa and elevated the maltase activity in the ileal mucosa ($p < 0.05$). Ferrous sulfate (FS) supplementation had no significant influence on maltase activity in the duodenum and jejunum (Table 5). However, FG supplementation elevated the maltase activity in the duodenum and jejunum ($p < 0.05$). Moreover, FG supplementation elevated the lactase activity in the ileal mucosa ($p < 0.05$).

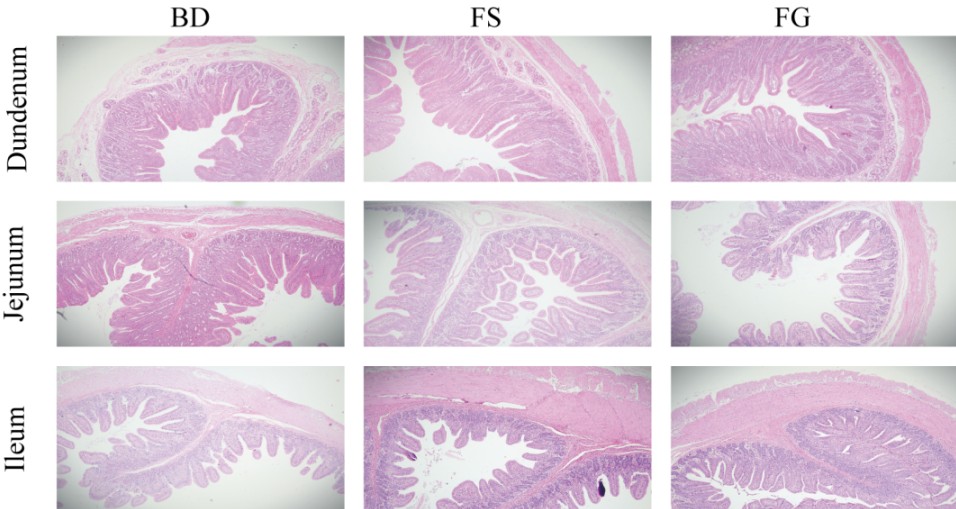

**Figure 1.** Effect of SPCI on intestinal morphology in weaned pigs (H&E; ×40). BD, basal diet with no iron supplementation; FS, basal diet with FeSO$_4$ (containing 100 mg Fe/kg) supplementation; FG, basal diet with Fe-Gly (containing 100 mg Fe/kg) supplementation.

**Table 4.** Effect of different iron sources on intestinal morphology in weaned pigs.

| ITEM | Treatments | | | SEM | *p*-Value |
|---|---|---|---|---|---|
| | **BD** | **FS** | **FG** | | |
| Duodenum | | | | | |
| Villus height (μm) | 361.99 [b] | 451.65 [a] | 498.59 [a] | 16.79 | 0.00 |
| Crypt depth (μm) | 256.44 [a] | 213.74 [ab] | 177.61 [b] | 11.19 | 0.01 |
| V/C | 1.45 [c] | 2.18 [b] | 2.84 [a] | 0.14 | 0.00 |
| Jejunum | | | | | |
| Villus height (μm) | 370.19 | 408.96 | 402.04 | 9.77 | 0.24 |
| Crypt depth (μm) | 220.59 [ab] | 230.47 [a] | 190.33 [b] | 7.92 | 0.09 |
| V/C | 1.75 | 1.80 | 2.14 | 0.08 | 0.11 |
| Ileum | | | | | |
| Villus height (μm) | 312.60 [b] | 339.44 [b] | 399.13 [a] | 10.57 | 0.00 |
| Crypt depth (μm) | 188.39 | 196.37 | 169.95 | 7.81 | 0.38 |
| V/C | 1.69 [b] | 1.78 [b] | 2.45 [a] | 0.11 | 0.00 |

V/C, villus height: crypt depth. Within a row, means without a common lowercase superscript differ ($p < 0.05$). Mean and total SEM are listed in separate columns (n = 8). BD, basal diet with no iron supplementation; FS, basal diet with $FeSO_4$ (containing 100 mg Fe/kg) supplementation; FG, basal diet with Fe-Gly (containing 100 mg Fe/kg) supplementation.

**Table 5.** Effect of different iron sources on mucosal enzyme activity in weaned pigs.

| ITEM | Treatments | | | SEM | *p*-Value |
|---|---|---|---|---|---|
| | **BD** | **FS** | **FG** | | |
| Duodenum | | | | | |
| Lactase (U/mgprot) | 2.19 | 3.45 | 4.08 | 0.39 | 0.12 |
| Sucrase (U/mgprot) | 11.85 | 10.43 | 11.71 | 0.71 | 0.69 |
| Maltase (U/mgprot) | 12.57 [b] | 14.90 [ab] | 17.80 [a] | 0.84 | 0.03 |
| Jejunum | | | | | |
| Lactase (U/mgprot) | 44.23 | 34.51 | 39.52 | 2.91 | 0.40 |
| Sucrase (U/mgprot) | 4.67 [c] | 10.38 [b] | 19.41 [a] | 1.52 | 0.00 |
| Maltase (U/mgprot) | 40.58 [b] | 34.16 [b] | 55.32 [a] | 3.64 | 0.04 |
| Ileum | | | | | |
| Lactase (U/mgprot) | 1.22 [b] | 1.85 [ab] | 2.51 [a] | 0.23 | 0.06 |
| Sucrase (U/mgprot) | 22.90 | 28.40 | 29.92 | 2.05 | 0.36 |
| Maltase (U/mgprot) | 23.96 [b] | 48.80 [a] | 48.27 [a] | 4.04 | 0.01 |

Within a row, means without a common lowercase superscript differ ($p < 0.05$). Mean and total SEM are listed in separate columns (n = 8). BD, basal diet with no iron supplementation; FS, basal diet with $FeSO_4$ (containing 100 mg Fe/kg) supplementation; FG, basal diet with Fe-Gly (containing 100 mg Fe/kg) supplementation.

### 3.5. Effect of Different Iron Sources on ZO-1 Distribution and Apoptosis of the Intestinal Epithelial Cells

As compared to the BD and FS groups, the localization of tight-junction protein ZO-1 in the jejunum was improved in the FG group ($p < 0.05$). Ferrous glycine (FG) and ferrous sulfate (FS) supplementation had no influence on the early apoptosis rate of the cells in the jejunal epithelium (Figure 2). However, the rate of apoptosis, except for early, was increased in the FS group when compared to the BD and FG groups ($p < 0.05$).

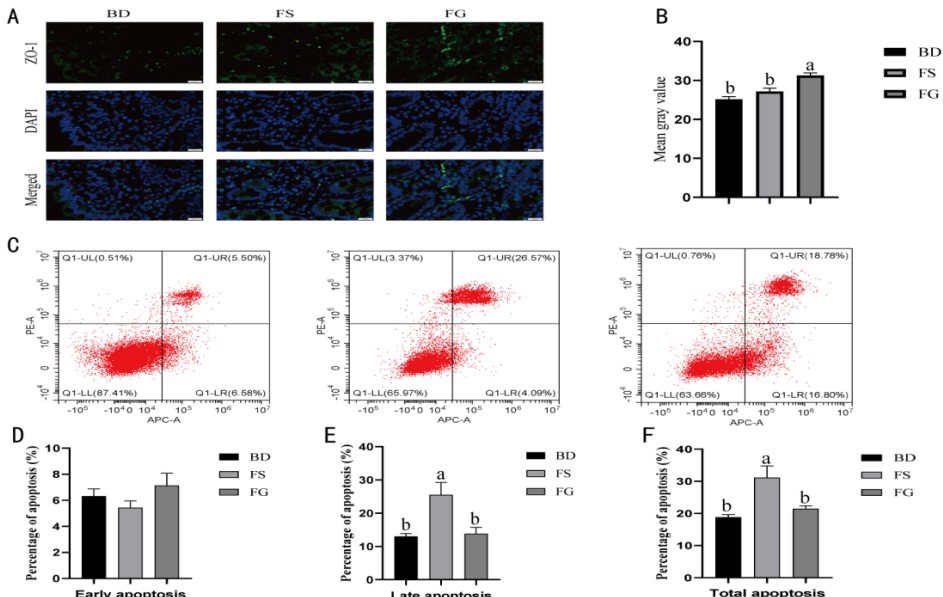

**Figure 2.** Effect of SPIC on ZO-1 distribution and apoptosis of the intestinal epithelium cells. (**A**) Localization of ZO-1 and DAPI (DNA) in the jejunum. ZO-1 protein (green), DAPI stain (blue), as well as merged ZO-1 protein and DAPI, are presented; (**B**) the mean gray value of ZO-1; (**C**) the result of jejunal cell apoptosis by flow cytometry in weaned piglets. A total of 30,000 cells were used in each acquisition reading. Frames were divided into 4 quadrants: Q1-UL represents necrotic cells; Q1-UR represents late apoptotic and early necrotic cells; Q1-LR represents early apoptotic cells; and Q1-LL represents normal cells; percentages of apoptotic cells of early apoptosis (**D**), late apoptosis (**E**), total apoptosis (**F**) in the jejunum, respectively. a,b,c Mean values with different letters on vertical bars indicate significant differences ($p < 0.05$). BD, basal diet with no iron supplementation; FS, basal diet with FeSO$_4$ (containing 100 mg Fe/kg) supplementation; FG, basal diet with Fe-Gly (containing 100 mg Fe/kg) supplementation.

### 3.6. Effect of Different Iron Sources on the Mucosal Antioxidant Capacity

As shown in Table 6, both FG and FS supplementation reduced the MDA content in the jejunal and ileal mucosa ($p < 0.05$). Ferrous glycine (FG) supplementation also elevated the activity of T-AOC in the jejunal mucosa and elevated the activities of T-AOC and CAT in the ileal mucosa ($p < 0.05$).

**Table 6.** Effect of different sources of iron on intestinal antioxidant in weaned pigs.

| ITEM | Treatments | | | SEM | *p*-Value |
|---|---|---|---|---|---|
| | BD | FS | FG | | |
| **Duodenum** | | | | | |
| GSH-Px (mg/mgprot) | 65.00 | 66.57 | 58.62 | 3.27 | 0.63 |
| T-AOC (U/mgprot) | 0.43 | 0.42 | 0.33 | 0.03 | 0.44 |
| MDA (nmol/mgprot) | 0.32 | 0.35 | 0.40 | 0.02 | 0.22 |
| CAT (U/mgprot) | 6.06 | 5.39 | 4.10 | 0.49 | 0.25 |
| T-SOD (U/mgprot) | 123.63 | 115.33 | 117.69 | 4.23 | 0.72 |
| **Jejunum** | | | | | |
| GSH-Px (mg/gprot) | 73.53 | 77.82 | 78.68 | 3.10 | 0.78 |
| T-AOC (U/mgprot) | 0.40 [b] | 0.52 [ab] | 0.69 [a] | 0.05 | 0.04 |
| MDA (nmol/mgprot) | 3.19 [a] | 2.30 [b] | 1.90 [b] | 0.18 | 0.00 |
| CAT (U/mgprot) | 13.95 [b] | 13.90 [b] | 19.00 [a] | 0.84 | 0.01 |
| T-SOD (U/mgprot) | 69.95 | 64.52 | 70.43 | 2.57 | 0.59 |
| **Ileum** | | | | | |
| GSH-Px (mg/gprot) | 52.81 | 57.55 | 50.29 | 2.56 | 0.51 |

**Table 6.** *Cont.*

| ITEM | Treatments | | | SEM | *p*-Value |
|---|---|---|---|---|---|
| | BD | FS | FG | | |
| T-AOC (U/mgprot) | 0.12 [b] | 0.14 [b] | 0.29 [a] | 0.03 | 0.01 |
| MDA (nmol/mgprot) | 0.50 [a] | 0.31 [b] | 0.34 [b] | 0.03 | 0.01 |
| CAT (U/mgprot) | 15.86 [ab] | 14.38 [b] | 21.28 [a] | 1.24 | 0.06 |
| T-SOD (U/mgprot) | 90.62 | 94.96 | 96.22 | 2.36 | 0.66 |

GSH-Px, glutathione peroxidase; T-AOC, total antioxidant capacity, MDA, malondialdehyde; CAT, catalase; T-SOD, total superoxide dismutase. Within a row, mean values of different letter superscripts were significantly different ($p < 0.05$). Mean and total SEM are listed in separate columns (n = 8). BD, basal diet with no iron supplementation; FS, basal diet with $FeSO_4$ (containing 100 mg Fe/kg) supplementation; FG, basal diet with Fe-Gly (containing 100 mg Fe/kg) supplementation.

### 3.7. Effect of Different Iron Sources on the Expression Levels of Critical Genes Related to Nutrient Digestion and Absorption

As shown in Figure 3, FG supplementation significantly elevated the expression levels of CAT1 in the duodenal and jejunal mucosa ($p < 0.05$). Moreover, the expression levels of *GLUT2* in the duodenal and ileal mucosa, as well as the expression level of *SGLT1* in the ileal mucosa, were elevated with FG supplementation ($p < 0.05$).

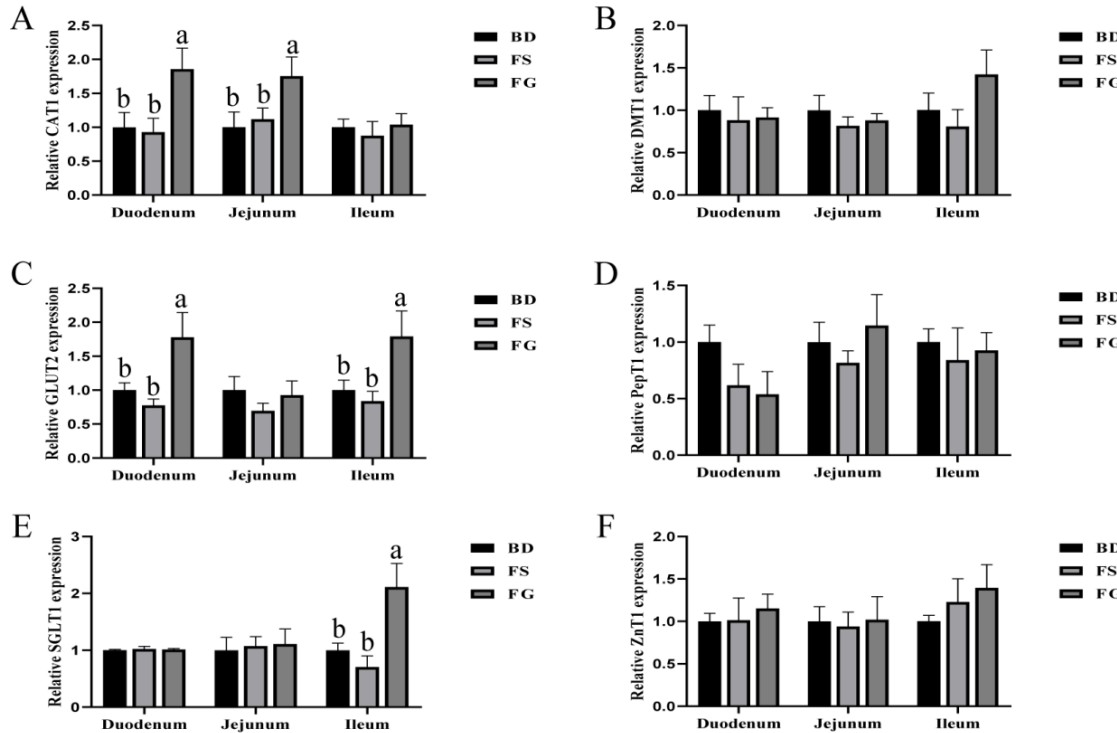

**Figure 3.** Effect of SPCI on the expression levels of critical genes related to nutrient digestion and absorption. *CAT1*, amino acid transporter-1 (**A**); *DMT1*, divalent metal transporter-1 (**B**); *GLUT2*, glucose transporter-2 (**C**); *PePT1*, peptide transporter-1 (**D**); *SGLT1*, sodium glucose transport protein-1 (**E**); *ZnT1*, zinc transporter-1 (**F**); Within a row, means without a common lowercase superscript differ ($p < 0.05$). BD, basal diet with no iron supplementation; FS, basal diet with $FeSO_4$ (containing 100 mg Fe/kg) supplementation; FG, basal diet with Fe-Gly (containing 100 mg Fe/kg) supplementation.

## 4. Discussion

For mammalian animals, including pigs, intestinal iron absorption from the diet is the major avenue to obtaining sufficient iron after weaning. Iron deficiency in animals not only decreases growth performance but also impairs erythropoiesis and immunity in a variety of animal species [5,28–30]. To prevent iron deficiency, it is a common practice to supply

the diet with various iron supplements [31]. Ferrous glycine (FG) is an organic form of iron that has been reported to be more palatable and effective than those inorganic irons (e.g., ferrous sulfate) [32,33]. In the present study, FG supplementation significantly decreased the F:G ratio in the weaned pigs, indicating a suitable effect of FG addition on growth performance. Previous studies indicated that dietary organic iron (e.g., amino acid chelated iron) supplementation can promote iron deposition in tissues [34]. In the present study, FG supplementation increased the iron deposition in the liver and tibia, which indicated a higher bioavailability compared to FS.

The storage of iron in the body can be evaluated based on several blood indicators, such as the HGB, RBC, and HCT, and iron deficiency will lead to a reduction in these parameters [35]. Ferritin is a primary iron storage protein that acts as a buffer pool against iron deficiency and iron overload [36]. In the present study, FG and FS supplementation not only elevated the content of HGB, RBC, and HCT but also significantly elevated the serum concentration of ferritin, indicating the necessity of iron supplementation in the weaning diet of pigs. These results are also consistent with previous studies on weaned pigs [37]. Iron can also act as an indispensable nutrient for the development or maintenance of the immune system [38]. Immunoglobulins are a class of glycoproteins produced by plasma cells and are very important for the immune response [39]. Moreover, they were classified into five isotypes, including IgM, IgG, IgA, IgD, and IgE [40]. In these five categories, IgG is the primary immunoglobulin in serum, which can promote phagocytosis of mononuclear macrophages and neutralize the toxicity of bacterial toxins [41]. In contrast, during immune responses, IgM is the first to appear, and it not only serves as the first line of host defense against infections but also plays an important role in immune regulation and immunological tolerance [42]. In this study, FG supplementation significantly elevated the concentrations of serum IgG and IgM in weaned pigs, which suggested an improved immunity in weaned pigs. Moreover, the result is also consistent with previous studies on a variety of animal species [43,44].

The integrity of intestinal epithelium is closely related to the digestion and absorption of nutrients [45,46]. In the present study, FS and FG supplementation elevated the villus height and the ratio of V/C in the duodenum, and FG supplementation also elevated the villus height and the ratio of V/C in the ileum. Sucrase and lactase are two important disaccharide enzymes present in small intestinal brush border membranes that are involved in carbohydrate digestion [47]. Moreover, the activities of maltase and sucrase are critical markers for evaluating intestinal development [48,49]. In our study, with FG added to the diet, the activities of sucrase and maltase in the jejunum and ileum improved, respectively. The result is compatible with the studies conducted on piglets [50,51], and both results indicated improved digestion and absorption capacity in the intestinal epithelium upon FG supplementation. ZO-1 is associated with the maintenance and regulation of the intestinal epithelial barrier and barrier function and is involved in important processes such as cellular material transport and maintenance of epithelial polarity [52]. In this experiment, the abundance of ZO-1 in the jejunum epithelium improved by the addition of FG, which indicated improved integrity of the intestinal epithelium upon iron supplementation. The result is consistent with the serum concentrations of endotoxin and D-lactate. Endotoxin is a potent immune stimulator that induces inflammation and antagonizes protein synthesis and digestibility in pigs [53–55]. D-lactic acid is produced by gut microbes, and only a small amount can be detected in the blood [56]. However, previous studies have shown that when the intestinal mucosa is damaged, a large amount of D-lactic acid will be absorbed into the bloodstream [57,58]. A previous study indicated that iron can act as a producer of reactive oxygen species (ROS) through the Fenton reaction, which subsequently leads to overproduction of ROS, DNA damage, and intestinal cell apoptosis [8,59]. In the present study, the late and total apoptosis rates of the jejunal epithelial cells were higher in the FS group than in the BD group. In contrast, the apoptosis rate in the FG group was lower than that in the BD group, which may be partially attributed to glycine as it has been previously reported to improve the intestinal mucosal barrier function by enhancing

mucosal protein synthesis and reduce the apoptosis of the intestinal epithelial cells [60,61]. Moreover, glycine is a critical component of glutathione (GSH) [62], one of the most abundant intracellular antioxidants that protect cells from oxidative stress [63]. In the present study, FG supplementation elevated the antioxidative capacity (as indicated by increases in jejunal and ileal CAT and T-AOC) in the weaned piglets. The result is also consistent with previous studies on broilers [64,65]. Both results suggested that the organic irons, such as FG, may act as one of the "safe" iron sources for the weaned piglets.

We also determined the expression levels of critical genes related to nutrient transportation. The divalent metal transporter-1(DMT1) is the only known membrane transporter for the intestinal absorption of non-heme iron, primarily at the duodenum brush border [66]. In our study, we observed increased iron content in tissues without changing DMT1 expression. However, the expression level of CAT1 was only elevated by organic iron (FG), which is consistent with the elevated iron deposition in the FG group. Unlike FS, organic irons, such as Fe-Gly, can be co-transported into the intestinal epithelial cells via various amino acid transporters [67]. SGLT1 and GLUT2 are two major glucose transporters that are responsible for maintaining glucose homeostasis [68,69]. Compared to the BD and FS groups, the mRNA expression levels of *SGLT1* and *GLUT2* in the ileum were elevated in the FG group, indicating a beneficial effect for sugar absorption upon FG supplementation. CAT1 is a major transmembrane transporter for L-arginine in endothelial cells [70]. Importantly, L-arginine can serve as the sole substrate for NO generation that is involved in regulating epithelial barrier function [71]. In this study, FG significantly elevated the expression levels of CAT1 in the duodenum and jejunum, which further indicated a beneficial role of FG in regulating intestinal barrier functions.

## 5. Conclusions

In summary, our study showed a beneficial effect of iron supplementation on immunity and intestinal health in weaned pigs. As compared with inorganic irons such as FS, FG had a higher bioavailability, as reflected by elevated iron deposition and improved intestinal barrier functions. These attributes should make FG an attractive substitute for conventionally used iron sources.

**Supplementary Materials:** The following supporting information can be downloaded at: https://www.mdpi.com/article/10.3390/agriculture12101627/s1, Table S1. Composition and nutrient level of experimental diet. Table S2. Primers sequences used for quanti-tative RT-PCR.

**Author Contributions:** L.S.; Conceptualization, Supervision, Methodology, Funding acquisition, Writing—original draft, B.Y.; Visualization, Y.L., Investigation, Resources, P.Z., Data curation, Z.H., Validation, J.Y., Methodology, Software, X.M., Formal analysis, H.Y., data curation, S.W., Project administration, J.H., Writing—review and editing, Funding acquisition. All authors have read and agreed to the published version of the manuscript.

**Funding:** This study was supported by the Key Research and Development Program of Sichuan Province (no. 2020YFN0147) and the National Natural Science Foundation of China (no. 31972599).

**Institutional Review Board Statement:** All experimental protocols used in the animal experiment were approved by the Institutional Animal Care and Use Committee of Sichuan Agricultural University (no. 20181105).

**Data Availability Statement:** The data used to support the findings of this study are available from the corresponding author upon request.

**Acknowledgments:** We thank Jiangsu Shuxing Biotechnology Co., Ltd. for their generous donation. We would also like to acknowledge and express our appreciation for Wang Huifen and Wu Fali for their help during the animal trial and sample collections.

**Conflicts of Interest:** The authors declare that there is no conflict of interest.

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
