# Peer review of "Effects of Different Sources of Iron on Growth Performance, Immunity, and Intestinal Barrier Functions in Weaned Pigs"

_agriculture, doi:10.3390/agriculture12101627_

Round 1

Reviewer 1 Report

Sun et al., investigated the effect of iron sources on pig growth performance and health status after weaning. Overall, it is a well-written manuscript. However, I still have suggestions and concerns before acceptance. The questions are listed below:

If you describe gene expression please use Italic. For example, GLUT2 should be changed to "GLUT2". Please go over the manuscript and change accordingly. 

Introduction

As you emphasized the role of iron in regulating oxidative and gut health. Please stress the importance of oxidative stress and gut health for pigs after weaning in the introduction. For example, how does oxidative stress and gut health impairs growth performance and immunity. You should list more references to clarify it. You should also clarify the logic that why iron supply contributes to reduced oxidative stress. To my knowledge, increased iron actually enhances oxidative stress via ferroptosis. 

Line36-38 references should be included. 

Line47-48 references should be added.

Materials and Methods

Both iron deficiency and overload impairs health. Therefore, how did select the dose of iron added to the diet? In other words, why do you think this dose is beneficial? Moreover, is it added based on the weight of diet (kg diet) or the weight of animals (kg animal weight)? Please clarify. 

How did you add the iron to the basal diet? Are they mixed with the basal diet or did you top dressed iron on the basal diet daily?

Are the pigs used in this experiment male or female? Are pigs allocated to each treatments from the same littermates? Do the have an adaptive period to the treatment diets? 

Did you collect blood after 12h fasting?

Results and discussion 

I did not see the result of iron content in fecal samples in Table 1? Not sure if I missed it. If you missed it, please add it. 

Final body weight and feed intake should be provided in Table 1.

You did not measure blood glucose; however, you determined the gene expression of glucose transporters. I did not see the logic of testing the gene expression of glucose transporters. Please clarify. Besides, you did not see any differences in iron transporters among different treatments. Why did you observed increased iron levels in tissues with Fe supplementation. Please clarify in the discussion. Same question for CAT1. You did not measure Arg. 

You should not only explain the differences you observed. Some insignificant results or the results are contrary to previous studies should be discussed as well. For example, why iron supply increases iron content in tissues without changing gene expression of iron transporters?

More discussion and references should be provided to support the result of FS supply increases apoptosis. As you mentioned in the introduction iron supply should be good for pigs regardless of its resources. While you observed enhanced apoptosis? Please compare it with previous studies and summarize it in the discussion section. 

Line340 delete "."

Reviewer 2 Report

The authors have assessed the effects of different iron sources on growth performance, immunity, and intestinal barrier functions in weaned pigs. This study evaluated several parameters related to serum, feces, and intestines. The findings are interesting and can be used in the future to help improve swine nutrition.

 Overall, the study is well designed, clearly presented, and uses appropriate methodologies for different objectives. The manuscript contains significant errors in writing English. Throughout, there are many instances of singular and plural not matching and sentences that are not organized correctly. The reviewer will not point out every instance; authors need to check carefully.

There are some issues that need to be considered to further improve the quality of the manuscript before publication.

Comments:

L 21/54: Do not use abbreviations at the beginning of the sentence. Correct throughout the text.

L 24 : FG  >> FG supplementation/group.

L 27-29: There was no improvement in the growth performance.

L  56: A previous study/Previous studies.

L 70-76: Animal information is too superficial. What sex? Male or female? The ratio of male/female?

M&M: The presented information does not highlight FS and FG were administered. How the supplementation dose was selected? Reference?

Pigs were fed how many times per day? Which times?

L 78: “ad libitum” should be in italic.

L 88: -20°C >> -20 °C; the writing format should be consistent throughout the text.

L 96: From which location?

L 166: **?

Statistical analysis: The Duncan test should not be used for multiple means comparisons. Although it name suggests differently, this test does not take the repetition on tests into account that occur when you compared more than two means against each other. However, this must be done to correct the threshold of significance because a higher repetition of tests increases the probability to detect a false-positive result just by chance. Hence, this dataset should be revised applying more suitable post-hoc tests, e.g., Tukey or Student-Newman-Keuls Test. Consult the following open-access source for more details. I also added the link where you can access the respective chapter on multiple comparisons within this e-book.

McDonald, J. H. 2014. Handbook of Biological Statistics, 3rd ed. Sparky House Publishing, Baltimore, MD, USA. (http://www.biostathandbook.com/multiplecomparisons.html)

Results: The authors should avoid “As shown in Table/Figure” at the beginning of the paragraph; use some introductory sentences.

As the significant level (P < 0.05) is indicated in the description of the results; it is recommended to delete the "significantly" from the text.

The format of P < 0.05 should be consistent throughout the text.

L 228: Define “IBW” at its first appearance in the text.

L 229: among

L 230: groups

L 230-232: No data for fecal sample in Table 1.

L 232-233: Compared with which group?

L 237-240: Superscript 1, 2, and 3 should be indicated in Table. Correct for all Tables.

L 266-270: It is recommended to describe in the sense of morphological structure (Figure 1). The villus height and crypt depth results are already presented in Table 4.

L 294: groups.

L 333-335: What do mean by the iron deficiency in the diet? Rephrase this sentence for better clarification.

L 337 : e.g.,

L 338: FG supplementation.

Conclusions should be improved; as growth performance was not improved. In addition, authors should indicate/suggest the more effective supplement for weaned pigs.

Reviewer 3 Report

Dear Editor,

The article is entitled "Effects of different sources of iron on growth performance, immunity, and intestinal barrier functions in weaned pigs" has been reviewed. The work has some value, but the data points aren't very good and the flow of information isn't smooth. The authors will need to overhaul and revise the manuscript completely and improve the presentation. There is a lack of information on the trial, specifically on diet formulation or containment of iron. Is supplementation of iron an overdose or a deficiency in the diets?

The authors have used iron supplementation in the diet of pigs in two forms of iron: either organic or inorganic at a level of 100 mg/kg feed. However, I would like to clarify that the pigs require 80–100 mg of iron for each kg of feed based on recommendations by NRC (2012). This study has not indicated the iron concentration in the diet or the diet ingredients. Some feedstuffs already contain iron, as well as a 2% premix in diets that also has iron. The authors targeted different health indicators related to the study objective, but none of these parameters were linked to the deficiency of iron since there was no negative treatment in this study. I don't think that any of the nutrients in the formulated diet (basal diet) should be missing. Therefore, I suggest to reject it.

Regards

Round 2

Reviewer 2 Report

The manuscript has been improved and addressed all of the concerns. It can be accepted for publication in its current form. However, the English language should be further revised by an expert for better readability.

Reviewer 3 Report

The effects of utilizing various iron sources on the growth performance and intestinal health of weaned pigs are of interest to the readership. The paper does not require any additional revisions. Therefore, I suggest you accept it.